# Associations between significant head injury in male juveniles in prison in Scotland UK and cognitive function, disability and crime: A cross sectional study

**T. M. McMillan**[1]*, **Julia McVean**[1], **Hira Aslam**[1], **Sarah J. E. Barry**[2]

**1** Institute of Health and Wellbeing, University of Glasgow, Gartnavel Royal Hospital, Glasgow, United Kingdom, **2** Department of Mathematics and Statistics, University of Strathclyde, Glasgow, United Kingdom

* thomas.mcmillan@glasgow.ac.uk

## Abstract

### Background

Although the prevalence of head injury is estimated to be high in juveniles in prison, the extent of persisting disability is unknown and relationships with offending uncertain. This limited understanding makes it difficult to develop effective management strategies and interventions to improve health or reduce recidivism. This study investigates effects of significant head injury (SHI) on cognitive function, disability and offending in juvenile prisoners, and considers relationships with common comorbidities.

### Methods

This cross-sectional study recruited male juvenile prisoners in Scotland from Her Majesty's Young Offenders Institute (HMYOI) Polmont (detaining approximately 305 of 310 male juveniles in prison in Scotland). To be included juveniles had to be 16 years or older, fluent in English, able to participate in assessment, provide informed consent and not have a severe acute disorder of cognition or communication. Head injury, cognition, disability, history of abuse, mental health and problematic substance use were assessed by interview and questionnaire.

### Results

We recruited 103 (34%) of 305 juvenile males in HMYOI Polmont. The sample was demographically representative of juvenile males in prisons for young offenders in Scotland. SHI was found in 82/103 (80%) and head injury repeated over long periods of time in 69/82 (85%). Disability was associated with SHI in 11/82 (13%) and was significantly associated with mental health problems, particularly anxiety. Group differences on cognitive tests were not found. However the SHI group reported poorer behavioural control on the Dysexecutive Questionnaire and were more often reported for incidents in prison than those without SHI. Characteristics of offending, including violence, did not differ between groups.

**Data Availability Statement:** Data cannot be shared publicly because of restrictions of the ethics board. Application can be made to Research

Governance at the University of Glasgow for permission to access the data; research-governance@glasgow.ac.uk

**Funding:** TMcM Scottish Government The funders had no role in study design, data collection and analysis, decision to publish, or preparation of the manuscript.

**Competing interests:** The authors have declared that no competing interests exist.

## Conclusions

Although SHI is highly prevalent in juvenile prisoners, associated disability was relatively uncommon. There was no evidence for differences in cognitive test performance or offending in juveniles with and without SHI. However, signs of poorer behavioural control and greater psychological distress in juveniles with SHI suggest that they may be at greater risk of recidivism and of potentially becoming lifelong offenders. This implies a need for remedial programmes for juvenile prisoners to take account of persisting effects of SHI on mental health and self-control and education and to improve their understanding of the effects of SHI reduce the likelihood of cumulative effects from further SHI.

## Introduction

Attention has been drawn to the burden of poor health in vulnerable and marginalised social groups and the need for effective policies and services that will reduce social and health inequalities [1]. Prisoners are one such group. They often have a background of social and economic deprivation and early life adversity, factors that are associated with health problems and antisocial behaviour including in juveniles [2]. Deficits in self-control are central to many theories of crime. Neurobiological reward systems that are not well controlled can cause a failure to assess risk and behaviours that are associated with sensation seeking and rule breaking. This risk is heightened in adolescence where there is an imbalance between well developed reward systems that respond to emotion and self-regulatory systems that are cognitively mediated and that are continuing to mature [3–5].

Arguably, neurodevelopmental challenges or environmental adversity could exaggerate an imbalance of this kind in adolescence or prolong it into adulthood. Consistent with this, adolescents who offend are at greater risk of having neurodevelopmental, mental health and substance misuse problems with associated distress, depression, self-harm and PTSD, and are characterised by poorer neuropsychological test performance, poorer self-control, negative emotions and risk taking behaviour [2, 3]. These characteristics of juvenile offenders are common after head injury, as also is poor self-monitoring, poor judgement and a lack of empathy, and they can result in repeated risk taking, rule breaking, antisocial behaviour and violence [4]. Indeed, head injuries are more often found in offenders than in the general public [6] and there is growing concern about the long term impact of head injury in young offenders [7, 8]. It is known that the brain continues to mature into the mid 20s and the effects of damage earlier in life may become increasingly apparent in adolescence, when there are complex demands associated with social judgement. Head injury early in life is associated with poorer adult outcomes including in educational attainment and health [9]. More specifically head injury has the potential to affect normal brain development and have neurobehavioural effects that include poorer impulse control [7, 10]. Given this a causative link between head injury, crime and recidivism is plausible, particularly given the high prevalence of head injury in juvenile offenders [11] and indeed there is some evidence to support this [12, 13]. There is a need however, to account for the contributions of co-morbidities that are associated with deprived backgrounds and not only to investigate whether there is a history of head injury, but if the severity is likely to have caused brain damage and persisting disability. There is little published evidence in this area and none on disability, except in adult female prisoners [14, 15]. An understanding of the potential contribution of head injury to offending is important when

developing services and determining what assessments and inputs are required and when [14, 16]. Hence, this study investigates relationships between a history of significant head injury (SHI) and other current and historical factors, and the outcomes of cognitive function, disability and offending. We use the term offender in this study for clarity and in accordance with current usage in the criminal justice and forensic mental health systems, and it is not intended to be pejorative.

## Methods

The study took place at HM Young Offenders Institute (HMYOI) Polmont which is Scotland's national holding facility for male young offenders aged 16–21 years. There are two other institutions in Scotland, both of which take adults and juveniles: HMP and YOI Grampian and HMP and YOI Cornton Vale. The last takes women only. As the number of juvenile females in prison in Scotland is small and was about 15 during the study period [17], the study only included males, given potential gender differences in causes of head injury and comorbidity [14]. The average number of juvenile males in prison in Scotland during the recruitment period was 310 of which 98% were resident in HMYOI Polmont [17]. Juvenile males were recruited from Polmont between January 17, 2019 and August 30, 2019. To be included, participants had to be aged 16 years or older, fluent in English, not have a severe acute disorder of cognition or communication and give informed consent. Parental consent was not a requirement of the Ethics Committee and was not obtained. They were recruited through word of mouth and posters placed in prison halls. To avoid bias towards recruiting participants with head injury, the project was advertised as a study on prisoner health. Those with interest were to ask a prison officer for an information sheet and if wishing to take part, a prison officer informed the researchers. Potential prisoner participants then met with a researcher (JM or HA) to find out more about the study and provide written consent. It was made clear to potential participants that they could withdraw from the study at any time and did not need to give an explanation. It was also made clear to them that taking part in the study would not affect any treatment they may be undergoing or their custodial sentence. If a researcher became aware of a significant health issue for a participant, this information was passed to prison health care staff with the participant's consent. If a researcher became concerned about a participant's risk to self or others this information would be passed on to the relevant prison officer who could signpost relevant agencies and if deemed necessary, this action would be taken without the participant's consent.

Prisoners are allocated a personal officer who has a case management role and meets them regularly. They encourage prisoner engagement including in rehabilitation or reintegration initiatives, and update case records and reports. The personal officer for each recruited prisoner was asked to complete a subset of independent observer forms about the prisoner, and commented on level of functioning and behavioural/management challenges.

## Procedure

Assessments took 1–2 hours with breaks if necessary and were carried out by a final year doctorate in clinical psychology trainee (JMcV) or by an experienced research worker (HA). Details of assessor training are given here [S1 File]. If the participant was fatigued or seemed unwell, as indicated by self-report or by observation of the assessor, the assessment was completed in a second session.

## Measures

Participants completed a demographic information questionnaire that also included questions about schooling, history of drug and alcohol use and offence history. Categories of seriousness of violent offending were based on Williams et al [18] [S2 File]. Deprivation quintiles were derived from postcodes using the Scottish Index of Multiple Deprivation (SIMD) [19]. SIMD ranks deprivation in Scotland across 6976 small geographical areas and considers income, employment, education, health, access to services, crime, and housing.

**Head injury:** The Ohio State University Traumatic Brain Injury Identification Method (OSU-TBI) is a structured interview for the assessment of history of head injury. It is valid for prison samples [20] and practical to use in prisons in Scotland [15]. The OSU-TBI records information on cause and severity of single-event and multiple head injury.

**Other CNS damage.** A separate form associated with the OSU-TBI was administered which asks about a history of other CNS diagnoses [20]. It was also noted if a participant indicated that they had lost consciousness for a cause other than head injury (eg cerebral anoxia) during administration of the OSU-TBI.

**Cognitive function.** Four tests were given to assess processing speed, attention and visual scanning (Symbol Digit Modalities Test) [21], auditory verbal list learning [22], processing speed and mental flexibility (Trail Making Test) [23] and verbal fluency (Benton's Controlled Oral Word Association Task) [24]. The Dysexecutive Questionnaire (DEX) [25] was completed by prisoners and the informant version by personal officers. The DEX assesses emotional, behavioural and cognitive difficulties associated with dysexecutive syndrome by self-report. Finally, the Word Memory Test assessed effort during cognitive test performance with a score below 34 at delayed recall suggesting poor effort [26].

**Disability.** The Glasgow Outcome at Discharge Scale (GODS) is a standardised, structured assessment of disability developed from the Glasgow Outcome Scale-Extended for use when individuals are not in the community [27]. Disability was rated as attributable to head injury or from any cause. The wording (but not structure) was altered to suit a prison context [15]. Cross-referencing with other information such as that obtained from the OSU-TBI, can be used to prompt during the interview and to assist in judging outcomes.

**Psychological distress and trauma.** The Hospital Anxiety and Depression Scale (HADS) was used to assess anxiety and depression by self-report with scores above 10 indicating moderate-severe caseness [28]. The Traumatic Life Events Questionnaire (TLEQ) [29] and the PTSD Checklist for the DSM-5 (PCL-5) [30] were used to assess lifetime exposure to traumatic events and PTSD respectively. Both have been used successfully with Scottish prisoners [15]. A score above 32 and fulfilment of criteria for intrusion, avoidance and hypervigilance on the PCL-5 suggests PTSD [31].The Adverse Childhood Experiences questionnaire (ACE) [32] assesses exposure to adverse life experiences before age 18.

**Alcohol and drug use.** The Drug Abuse Screening Test (DAST-10) [33] and the screening version of the Alcohol Use Disorders Identification Test Consumption (AUDIT-C) [34] were used to assess drug and alcohol use respectively. Scores greater than 5 for the DAST-10 and greater than 3 for the AUDIT-C categorised problematic use.

## Definition of groups

Offenders were grouped as having SHI if reporting a mild head injury with loss of consciousness for less than 30 minutes or moderate-to-severe head injury with loss of consciousness for at least 30 minutes, or head injury without loss of consciousness on more than two occasions. Offenders were classified without SHI if reporting no history of head injury or mild head injury on fewer than three occasions without loss of consciousness (NoSHI group) [15, 35].

## Data analysis

Analyses were performed in R (v-4.1.0). Continuous variables were summarised using mean and standard deviation (SD) or median and interquartile range (IQR), depending on the variable distribution. Categorical variables were summarised using counts and percentages. Two-sided t-tests or Mann-Whitney tests were used as appropriate to assess differences between the head injury groups in participant characteristics measured as continuous variables, while Fisher's Exact test or the chi-squared test was used for categorical variables. Age, years of education and delayed Word Memory Test score were used to adjust cognitive test scores and an overall summary adjusted z-score was subsequently calculated. Disability outcomes were aggregated into disability (moderate/severe) vs no disability (NoSHI/good recovery/no disability/possible impairment).

A linear regression model was fitted to the continuous outcome (cognitive impairment continuous z-score), logistic regression models to the binary outcomes (disability and violent offending) and quasi-Poisson models to the count outcomes (number of convictions and longest sentence; allowing for overdispersion). Univariable models assessed the unadjusted differences between groups, while multivariable models adjusted for historical (any history of abuse from TLEQ; self-reported problematic drug or alcohol use; CNS diagnosis at any age) and current (PTSD diagnosis; current depression or anxiety from the HADS) characteristics. The model for SHI attributed disability used AUDIT and DAST scores to represent alcohol and drug misuse, respectively, due to zero cell counts for the categorical problematic use variables. Model estimates of the group differences are reported as mean difference (linear model), odds (logistic model) or rate (quasi-Poisson model) ratios with corresponding 95% confidence intervals. Model fit was assessed visually via residual plots and using the Hosmer-Lemeshow test for the logistic regression models [S2 File].

Pearson correlation coefficients were used to assess collinearity between explanatory variables, with the highest correlation of 0.4 being between current depression and current anxiety and PTSD diagnosis and all others being below 0.35. We considered these low enough to justify inclusion of these explanatory variables in the models.

## Ethics approvals

Permission was obtained from the West of Scotland NHS Research Ethics Committee (18/WS/0210) and from the Scottish Prison Service Research Ethics Committee (no reference; date 6 November 2018). Written consent was obtained from all participants. The ethics committees did not require that parental consent to take part was obtained.

## Results

The sample of n = 103 represents approximately a third of the juvenile male prison population in Scotland [13]. The average age of the sample was 19.0 years (range 16–22), with the majority aged 18 and over (84; 82%). Almost all (96/102; 94%) self-described as ethnically white. Four self-described as mixed-race or multi-race, one as Asian and one as Black. Three quarters (73/96; 76%) were from the two most socially deprived quintiles [19]. Scottish prison statistics for 2019–2020 [17] reveal very similar demographics to that of the sample for age (83% aged over 17 years) ethnicity (94% white) and social deprivation quintiles (45% most deprived quintile) suggesting that the sample was representative of the juvenile prison population (Table 1 and S2 File). More than half had received 1:1 support in school or special schooling (63/102; 62%). Almost all had truanted (94/98; 96%) with 74/98 (76%) doing so weekly and schooling being further disrupted by suspension or exclusion in the majority (84/96; 88%). Many were unemployed before imprisonment (44/102; 43%). Differences between SHI (n = 82) and NoSHI

**Table 1. Baseline characteristics.**

| Variable | Statistic | Total (N = 103) | SHI (N = 82) | NoSHI (N = 21) | P-value |
|---|---|---|---|---|---|
| Age (years) | $N_{obs}$ ($N_{miss}$) | 103 (0) | 82 (0) | 21 (0) | |
| | Mean (SD) | 19 (1) | 19 (1) | 19 (1) | |
| | Range | (16, 22) | (16, 22) | (17, 22) | 0.533 |
| Age band 16–17 | N (%) | 19 (18) | 15 (19%) | 4 (18%) | |
| >17 | N (%) | 84 (82) | 67 (81%) | 17 (82%) | 0.814 |
| Ethnicity | $N_{obs}$ ($N_{miss}$) | 102 (1) | 81 (1) | 21 (0) | |
| White | N (%) | 96 (94%) | 77 (95%) | 19 (90%) | |
| Non-white | N (%) | 6 (6%) | 4 (5%) | 2 (10%) | 0.600 |
| Scottish Index of Multiple Deprivation quintile | $N_{obs}$ ($N_{miss}$) | 96 (7) | 76 (6) | 20 (1) | |
| 1—most deprived | N (%) | 50 (52%) | 37 (49%) | 13 (65%) | |
| 2 | N (%) | 23 (24%) | 20 (26%) | 3 (15%) | |
| 3 | N (%) | 9 (9%) | 7 (9%) | 2 (10%) | |
| 4 | N (%) | 7 (7%) | 6 (8%) | 1 (5%) | |
| 5—least deprived | N (%) | 7 (7%) | 6 (8%) | 1 (5%) | 0.799 |
| Years of education | $N_{obs}$ ($N_{miss}$) | 102 (1) | 81 (1) | 21 (0) | |
| | Mean (SD) | 10 (1) | 10 (1) | 10 (2) | |
| | Range | (4, 13) | (7, 13) | (4, 12) | 0.739 |

groups (n = 21) were slight and not statistically significant for age, ethnicity, deprivation, years of education or school attendance (Table 1).

## Occurrence of head injury

The majority of participants reported a history of SHI (82/103; 80%). On the OSU-TBI, the 'worst' head injury was moderate (LoC 30 minutes to 24 hours) in 11/103 (11%) or severe (LoC >24 hours) in 1/103 (1%). For the remainder, the 'worst' head injury reported was mild, with LoC for less than 30 minutes in 47/103 (46%) and mild without LoC in 39/103 (38%). No history of head injury was reported in 5/103 (5%).

The SHI group (n = 82) comprised of the 12 juveniles with moderate or severe head injury (15%), 47 with mild head injury and LoC (57%) and 23 with more than two head injuries without LoC (28%; median 4, IQR 3–5). The median number of single incident SHI reported overall in the SHI group was 5 (IQR 3–7). All participants in the SHI group reported having symptoms after head injury. The majority of those in the SHI group (69/82; 85%) reported one or more periods of time where they sustained multiple head injury. The median duration of such episodes of repeat head injury was 5 years (IQR 3–8), and in 64/69 (82%) was for at least one year. Only one participant had a single incident moderate-severe head injury without multiple mild head injury in addition. In the SHI group the median age of the first head injury was 9 years (IQR 6–12) and the median age at first head injury with LoC, 15 years (IQR 12–17). Most of the SHI group reported not attending hospital after a head injury (61/77; 79%), with five being uncertain and duration of admission in the remainder ranging from less than one day to 75 days (IQR 1–9 days) [S2 File]. The NoSHI group (n = 21) comprised of the 5 participants with no history of head injury (24%) and 16 with fewer than three head injuries with no LoC (76%).

The lifetime causes of SHI were fighting or assault in 66/82 (80%), sporting activities which were most often boxing, football and cycling in 39/82 (48%), a fall in 37/82 (45%), a motor vehicle accident in 20/82 (24%) and other causes in 33/82 (40%). Relatively few in the SHI

group reported head injury resulting from parental or familial violence (14/82; 17%) or partner violence (7/82; 9%). Of the 69/82 (84%) with episodes of repeat HI, the most common causes were fighting/assault in 62/70 (74%) and sport in 30/70 (42%). These figures are based on the total number of head injuries or episodes reported in the SHI group, with many individuals having several. Twelve (12/103; 12%) reported having a mild head injury in recent weeks and one other a moderate HI. In the SHI group, age at first SHI was not significantly associated with SHI-related disability (OR 1.0, 95% CI 0.8–1.2; p = 0.785), overall cognitive function (difference in means -0.01, 95% CI -0.07 to 0.04; p = 0.625) or violent offending (OR 0.9, 95% CI 0.8–1.1; p = 0.179).

## Occurrence of other central nervous system diagnoses

A CNS diagnosis (other than HI) at any age was reported by 41/103 (40%); those with SHI (37/82; 45%) had a rate twice that of those in the NoSHI group (4/21; 19%; OR = 3.5, 95% CI 1.0–15.3; p = 0.044). A diagnosis of ADHD was self-reported by 29/103 (28%) with no differences in rate between SHI (25/82; 30%) and NoSHI (4/21; 19%) groups (OR = 1.85, 95% CI 0.5–8.3; p = 0.417). For other conditions the numbers were small, including Learning Disability (3/103; 3%), Autistic Spectrum Disorder (8/103;8%), cerebral anoxia through strangling (8/103; 8%) and epilepsy (4/103;4%) [S2 File].

## Physical and mental health

Physical problems not associated with recent drug use were rarely reported (4/103; 4%). These were migraine (1/103), pain (2/103) and nausea and dizziness not associated with head injury (1/103).

On the HADS, scores for anxiety and depression were both significantly higher in the SHI than in the NoSHI group (Table 2) and clinical anxiety more common in the SHI group (33/82;40%) than in the NoSHI group (0/21; 0%); p = 0.0001. No one in the NoSHI group was clinically anxious or depressed (Table 2). PCL-5 scores were significantly higher in the SHI

**Table 2. Mental health.**

| Variable | Statistic | Total (N = 103) | SHI (N = 82) | NoSHI (N = 21) | P-value |
|---|---|---|---|---|---|
| HADS depression score | $N_{obs}$ ($N_{miss}$) | 103 (0) | 82 (0) | 21 (0) | |
| | Median (IQR) | 5 [2, 8] | 5 [3, 9] | 2 [1, 4] | |
| | Range | (0, 17) | (0, 17) | (0, 7) | <0.001 |
| Clinical Depression (HADS depression>10) | $N_{obs}$ ($N_{miss}$) | 103 (0) | 82 (0) | 21 (0) | |
| | N (%) | 14 (14%) | 14 (17%) | 0 (0%) | 0.068 |
| HADS anxiety score | $N_{obs}$ ($N_{miss}$) | 103 (0) | 82 (0) | 21 (0) | |
| | Median (IQR) | 8 [5, 12] | 10 [6, 13] | 5 [3, 7] | |
| | Range | (0, 20) | (1, 20) | (0, 9) | <0.001 |
| Clinical Anxiety (HADS anxiety>10) | $N_{obs}$ ($N_{miss}$) | 103 (0) | 82 (0) | 21 (0) | |
| | N (%) | 33 (32%) | 33 (40%) | 0 (0%) | <0.001 |
| PCL-5 score | $N_{obs}$ ($N_{miss}$) | 103 (0) | 82 (0) | 21 (0) | |
| | Mean (SD) | 26 (19) | 29 (19) | 15 (15) | |
| | Range | (0, 73) | (0, 73) | (0, 46) | 0.001 |
| PTSD diagnosis | $N_{obs}$ ($N_{miss}$) | 103 (0) | 82 (0) | 21 (0) | |
| | N (%) | 32 (31%) | 29 (35%) | 3 (14%) | 0.070 |

group but differences became marginally non-significant (p = 0.070) when scores were converted using criteria for PTSD diagnosis, although the magnitude of difference in proportions was large (Table 2).

## Physical or sexual abuse

The total number of items endorsed on the TLEQ was significantly higher in the SHI group (median 7, IQR 5–9) than in the NoSHI group (median 4; IQR 3–5; p<0.001) suggesting exposure to a greater range of traumatic events [S2 File]. A third of juveniles reported physical punishment or sexual abuse before age 16 (33/103; 32%) and proportions were similar in each group (SHI 28/82, 34%; NoSHI 5/24, 24%; p = 0.440). Sexual abuse when aged 16 or older was relatively rare, occurring in 8/103. Being a victim of partner violence when aged 16 or older was reported by 43/103 (41%) and was more common in SHI (40/82, 49%) than in NoSHI groups (3/21, 14%; p = 0.015). However, few who reported partner violence associated these events with fear or helplessness (8/43, 19%) [S2 File].

Adverse childhood experiences (ACEs) were common (median 4; IQR 2–6) and similar in frequency in SHI (median 4; IQR 2–6) and NoSHI groups (median 3; IQR 1–6; p = 0.179). Only 6/103 participants reported no ACEs. More than half reported more than three ACEs (55/103; 55%), with group differences not statistically significant (SHI 47/82; 57%; NoSHI 8/21; 38%; OR = 2.2, 95% CI 0.7–6.7; p = 0.144). The ACE indicated that a history of parental incarceration was common (59/103, 57%) with no significant difference between groups (SHI 47/82, 57%; No-SHI 12/21, 57%; p = 1.000).

## Drug and alcohol misuse

Most participants self-reported a history of problematic alcohol or drug use (80/102; 78%). Although rates for both were higher in the SHI group, differences were statistically non-significant (alcohol OR = 2.5, 95% CI 0.8–8.8, p = 0.088; drugs OR = 2.9, 95% CI 0.96–9.0, p = 0.055; Table 3). Alcohol use was above the clinical cut-off on the AUDIT-C in 96/103 (93%) and drug use above the cut-off on the DAST in 43/103 (42%). Both were significantly more common in the SHI group (alcohol OR = 6.0, 95% CI 1.2–45.1, p = 0.030; drugs OR = 5.6, 95% CI 1.5–32.1, p = 0.006), although imprecisely estimated (Table 3).

**Table 3. Alcohol and drug use.**

| Variable | Statistic | Total (N = 103) | SHI (N = 82) | NoSHI (N = 21) | P-value |
|---|---|---|---|---|---|
| Problematic alcohol use | $N_{obs}$ ($N_{miss}$) | 102 (1) | 81 (1) | 21 (0) | |
| | N (%) | 47 (46%) | 41 (51%) | 6 (29%) | 0.088 |
| Problematic drug use | $N_{obs}$ ($N_{miss}$) | 102 (1) | 81 (1) | 21 (0) | |
| | N (%) | 73 (72%) | 62 (77%) | 11 (52%) | 0.055 |
| AUDIT-C total score | $N_{obs}$ ($N_{miss}$) | 103 (0) | 82 (0) | 21 (0) | |
| | Median (IQR) | 9 [6, 11] | 10 [7, 11] | 8 [5, 10] | |
| | Range | (0, 12) | (0, 12) | (0, 12) | 0.020 |
| AUDIT-C score>3 | $N_{obs}$ ($N_{miss}$) | 103 (0) | 82 (0) | 21 (0) | |
| | N (%) | 96 (93%) | 79 (96%) | 17 (81%) | 0.030 |
| DAST total score | $N_{obs}$ ($N_{miss}$) | 103 (0) | 82 (0) | 21 (0) | |
| | Median (IQR) | 5 [2, 7] | 5 [3, 7] | 3 [0, 5] | |
| | Range | (0, 10) | (0, 10) | (0, 9) | 0.008 |
| DAST score>5 | $N_{obs}$ ($N_{miss}$) | 103 (0) | 82 (0) | 21 (0) | |
| | N (%) | 43 (42%) | 40 (49%) | 3 (14%) | 0.006 |

## Disability

Disability that was attributed to SHI using the GODS was found in 11/82 (13%). Cause of disability was uncertain but possibly attributable to HI, in a further 1/82 (1%). Disability associated with SHI was moderate in 9/11 and severe in 2/11 (Table 4). Difficulties underpinning SHI-associated disability were most commonly mental health (10/11; 91%) and anger control (4/11; 36%). Two of the eleven juveniles who were disabled by SHI did not have disability from other causes.

In a multivariable logistic regression model, juveniles with current clinical anxiety were significantly more likely to have SHI-attributed disability (vs good recovery) than those without (OR 12.2, 95% CI 2.2–67.7), although this was imprecisely estimated due to the small number with SHI-attributed disability. SHI attributed disability was not significantly associated with current clinical depression, current PTSD, history of CNS diagnosis, abuse as a child or when aged 16 or over or alcohol or drug abuse assessed on the AUDIT-C or DAST [S3 File].

Disability from any cause was found in about half of participants on the GODS (53/103; 52%). SHI was associated with any cause disability in a univariable model (OR 3.5, 95% CI 1.2–10.0). When other potential risk factors were included as a multivariable model, SHI became non-significant when adjusting for current (OR 1.4; 95% CI 0.4–4.3) or historical factors (OR 2.6; 95% CI 0.9–7.8). In the multivariable model, current clinical anxiety was associated with any cause disability (OR 8.7, 95% CI 2.8–27.3). Current PTSD or clinical depression, a history of abuse, CNS diagnosis or alcohol or drug abuse, while all having positive ORs, did not show statistically significant associations with any cause disability [S3 File].

Disability was most commonly reported as resulting from mental health problems (73/103; 71%), particularly anxiety (28/103; 27%) and low mood (12/103; 12%). Problems resulting from anger and temper control were also often reported as disabling (36/103, 35%). Psychotic (largely paranoid) symptoms were reported in the context of effects of drug misuse (5/103; 5%). Group differences were not significant for any of these [S2 File].

## Cognitive function

Cognitive tests revealed little difference between SHI and NoSHI groups either as raw scores or after adjustment for age, years of education and delayed Word Memory Test score [S2 and S3 Files]. Correspondingly, the overall cognitive z-score showed little difference between groups (SHI mean -0.019; SD 0.989; NoSHI mean 0.073; SD 1.062) or after adjustment for historical (difference in means 0.20, 95% CI -0.30–0.69) or current risk factors (difference in

**Table 4. Disability outcome on the Glasgow outcome at discharge scale.**

|  | Statistic | SHI (N = 82) | NoSHI (N = 21) |
|---|---|---|---|
| **Disability from any cause** | $N_{obs}$ ($N_{miss}$) | 82 (0) | 21 (0) |
| Good recovery | N (%) | 32 (39%) | 14 (67%) |
| Moderate disability | N (%) | 34 (41%) | 5 (24%) |
| Severe disability | N (%) | 13 (16%) | 1 (5%) |
| No disability history | N (%) | 3 (4%) | 1 (5%) |
| **Disability from SHI** | $N_{obs}$ ($N_{miss}$) | 82 (0) | - |
| Good recovery | N (%) | 70 (85%) | - |
| Moderate disability | N (%) | 9 (11%) | - |
| Severe disability | N (%) | 2 (2%) | - |
| Unclear if head injury cause | N (%) | 1 (1%) | - |

means 0.20, 95% CI -0.30–0.71) [S3 File]. On the Word Memory Test 84/103 (82%) participants scored above the cut-off score of 33 at delayed recall, suggesting reasonable effort on cognitive tests. If excluding juveniles with Word Memory Test scores below 34 (SHI n = 18 and NoSHI n = 1), group differences for overall cognitive function remain small and non-significant (SHI mean -0.05; SD 0.90; NoSHI mean 0.12; SD 1.06; difference in means 0.21 95% CI -0.30, 0.72 adjusted for current risk factors; difference in means 0.09 95% CI -0.39,0.58 adjusted for historical risk factors).

Self-report scores on the DEX [S2 File] were significantly higher in the SHI group (mean 37.0, SD 15.0, n = 81) than in the NoSHI group (mean 26.0, SD 14.0, n = 21; 95% CI 3.4,17.4; p = 0.006). DEX scores did not differ by group for the independent report by prison officers (SHI mean 26.0, SD 17.0, n = 64; NoSHI mean 22.0, SD 15.0, n = 18; 95% CI -4.4,12.8; p = 0.356). DEX scores in the SHI group for 18 juvenile participants who were not independently rated (mean 37.5, SD 16.7), did not differ significantly from those who were independently rated (36.5, 15.3) by prison officers (p = 0.825).

## Offending

SHI and NoSHI groups did not differ in the number of convictions or longest length of sentence and the age at first offence was similar between groups (Table 5). There was a weak but statistically significant correlation between age at first SHI and age at first offence (r = 0.268; 95% CI 0.051–0.460; p = 0.016). Multivariable analysis indicated that a CNS diagnosis was marginally associated with having 1.9 times more convictions as those without (95% CI 1.0–3.7), while there was no evidence that SHI, any abuse, problematic alcohol or drug misuse, or current clinical anxiety, depression or PTSD was associated with number of convictions. Longest length of sentence or number of convictions was not associated with any current or historical factor in the multivariable analysis [S3 File].

Differences between SHI and NoSHI groups by type of offence, including violence, were not statistically significant (Table 5), A history of violent offences was very common (86/103; 84%). Differences in the most serious violent offence did not differ between groups (p = 0.435) [S2 File]. Multivariable analysis indicated that self-report of problematic alcohol or drug misuse was a risk factor for violent offending (OR 6.2, 95% CI 1.8–21.4), whereas a history of SHI, abuse, CNS diagnosis or current clinical anxiety, depression, PTSD or alcohol/drug misuse showed no evidence of association [S3 File]. More in the SHI group had recorded prison incidents (49/61; 80%) than in the NoSHI group (9/16; 56%; p = 0.047).

## Discussion

This cross-sectional study on one third of male juveniles in prison in Scotland, revealed a history of SHI in four out of five participants and in two thirds, mild head injury that had occurred repeatedly over lengthy periods of time. Despite the high prevalence, disability was associated with SHI only in one in eight, and cognitive test scores or characteristics of offending did not differ between groups with and without SHI. Those with SHI did more often have a history of other CNS diagnoses, clinical anxiety or depression and problematic alcohol or drug use. Of these factors, only clinical anxiety was associated with disability in the SHI group. There was an indication that the SHI group had greater difficulty with self-regulation and control of impulsive behaviour. This was evidenced by higher self-report scores on the DEX questionnaire and indications of behavioural expression of this in prison, where recorded prison incidents were more frequent in those with SHI. However, this effect is nuanced, given the absence of group differences in offending history, including violence.

**Table 5. Offending history.**

| Variable | Statistic | Total (N = 103) | SHI (N = 82) | NoSHI (N = 21) | P-value |
|---|---|---|---|---|---|
| Number of convictions | $N_{obs}$ ($N_{miss}$) | 102 (1) | 81 (1) | 21 (0) | |
| | Median (IQR) | 4 [2, 8] | 4 [2, 10] | 3 [1, 4] | |
| | Range | (0, 104) | (0, 90) | (0, 104) | 0.952 |
| Longest length of sentence (months) | $N_{obs}$ ($N_{miss}$) | 99 (4) | 78 (4) | 21 (0) | |
| | Median (IQR) | 10 [4, 19] | 10 [5, 18] | 9 [2, 22] | |
| | Range | (0, 48) | (0, 48) | (0, 42) | 0.745 |
| Age at first offence (years) | $N_{obs}$ ($N_{miss}$) | 101 (2) | 80 (2) | 21 (0) | |
| | Median (IQR) | 14 [12, 16] | 13 [11, 15] | 14 [12, 16] | |
| | Range | (7, 19) | (7, 18) | (9, 19) | 0.312 |
| Violent offences | $N_{obs}$ ($N_{miss}$) | 102 (1) | 81 (1) | 21 (0) | |
| | N (%) | 86 (84%) | 69 (85%) | 17 (81%) | 0.737 |
| Sexual offences | $N_{obs}$ ($N_{miss}$) | 102 (1) | 81 (1) | 21 (0) | |
| | N (%) | 14 (14%) | 11 (14%) | 3 (14%) | 1.000 |
| Property offences | $N_{obs}$ ($N_{miss}$) | 102 (1) | 81 (1) | 21 (0) | |
| | N (%) | 65 (64%) | 54 (67%) | 11 (52%) | 0.308 |
| Other offences | $N_{obs}$ ($N_{miss}$) | 102 (1) | 81 (1) | 21 (0) | |
| | N (%) | 57 (56%) | 48 (59%) | 9 (43%) | 0.220 |

Our sample was demographically similar to the juvenile prison population in Scotland suggesting it is representative. Juveniles in Scottish prisons most often have backgrounds of deprivation, disrupted schooling and unemployment or unskilled work and almost all identify as white [6, 17, 36]. As SHI and NoSHI groups did not differ on these factors, their backgrounds seem similar and group effects not likely to be artefacts of background differences.

Four in five juveniles in the present study had a history of SHI and this is much higher than the 30% reported in the meta-analysis by Farrer et al [11]. The disparity may partly be due to different definitions of head injury across studies. When studies on juvenile offenders define head injury as loss of consciousness and/or being dazed after head injury, prevalences are 67–72% and more consistent with the 80% found in the present study [37, 38]. When the definition is restricted to loss of consciousness, studies typically report a much lower prevalence (31–41%) [18, 39–41]. We believe it is important to include those with repeat head injury without loss of consciousness and symptomatic effects, given increasing evidence for cumulative and persisting effects of mild head injury [20, 35, 42].

Male juveniles with SHI in prison in Scotland differ from the general population with a history of head injury. Fighting and assault were by far the most common lifetime causes of SHI in juvenile prisoners (four in five), with a history of episodes of repeat head injury also typical. In contrast, a fall is the most common cause of hospitalised head injury in the general population and a history of single incident accidental head injury more typical [43, 44]. In the general population, head injury resulting from an assault is more likely in adolescents than in adults, although the prevalence is much lower than found in juvenile prisoners here [39, 40], with one review of eighteen studies indicating a median prevalence of 22% [45].

Studies on juvenile offenders [41, 46, 47] including our own, do not find poorer performance on cognitive tests in those with head injury. This may be because of widespread effects of deprivation and multiple morbidity on cognition in offenders [3, 40, 48]. Consistent with this, the overall sample reported here had significantly poorer scores on a test of mental flexibility than expected from general population norms [S2 File]. There is however evidence to

suggest that cognitive function is poorer in adult male prisoners with head injury than in those without [49]. This could imply that juvenile offenders with SHI who have poorer cognitive function more often continue offending into adult life or that juvenile offenders with SHI who continue to offend, are at greater risk of becoming cognitively impaired. Follow-up work is needed to elucidate this, given that intervention strategies would differ by aiming either to reduce the impact of existing cognitive impairment or reduce risk of future SHI.

Juveniles in prison often have neurodevelopmental difficulties, problems associated with mental health and substance misuse and previous trauma [2, 50]. Indeed, this was the case in our sample. However, few other studies have looked at relationships between these factors and head injury in juveniles. Similar to our own findings, those that have, report a greater risk of mental health problems including anxiety, trauma and substance misuse in those with head injury [39, 41, 46, 51]. In our study, group differences in those fulfilling diagnostic criteria for PTSD were statistically non-significant, although overall PCL-5 scores were significantly higher in the SHI group. This may reflect sample size limitations or alternatively, greater vulnerability to PTSD later in life in those with SHI. The higher scores on the TLEQ in the SHI group may support the latter interpretation. Clearly though, the higher PCL-5 and HADS-anxiety scores indicate greater distress in the SHI group and this is often associated with prisoners being more difficult to manage [52], as found in the SHI group who more often had recorded prison incidents.

There is very little evidence about disability after head injury in offenders [14] and none previously on disability after head injury in juvenile prisoners [S1 File]. The prevalence of disability after SHI found here (13%) was low compared to the very high prevalence of SHI and the 35–40% prevalence of SHI-attributed disability found in adult prisoners with SHI in Scotland [15, 53]. This difference between juvenile and adult prisoners might be explained by a continuing risk of developing SHI-associated disability in juveniles who continue to offend into adulthood. Cumulative damage from repeated mild SHI is known to occur in non-offender populations [20, 35, 42] and is a risk for juvenile prisoners if released into the same environment that previously exposed them to SHI. Education about the effects of head injury and support in the community might reduce this risk. Knowledge about head injury and its impact on behaviour is poor in adult male prisoners, but can be improved by brief education [54].

Other CNS disorders were not associated with disability in the overall sample, or in the SHI group. Clearly, diagnosis of CNS disorder is not synonymous with persisting disability. However for some, disability may be situation specific, occurring when higher demands are placed on functions that are weak or impaired, such as communication or memory during stressful legal processes. These difficulties may not be evident in normal daily routines [55], and the absence of an association between other CNS disorders and disability assessed on the GODS might reflect this.

Evidence for violent crime as a distinguishing characteristic in juvenile offenders with head injury is not strong, with most studies reporting no group effects between offenders with and without head injury [39, 40, 41, 56, 57]. This seems surprising, given evidence for greater impulsivity and reactive aggression after head injury in the general population, including in studies on exposed and non-exposed juvenile twins [58] and because theories of offending emphasise aggressiveness and poor self-control as being criminogenic [4, 58]. A few studies report greater risk of severe violence in juvenile detainees with head injury [18, 40] and Kenny et al [40] note the potential for head injury and drugs to act synergistically and reduce self-control. However other studies, including our own, find no greater risk of severe violence (or violence) in juvenile prisoners with head injury [38, 41]. Differences in environment, deprivation and exposure to head injury might explain these inconsistencies across studies. Indeed, evidence for head injury as a risk factor for violent offending is stronger in adult offenders [13]

where exposure to trauma, problematic alcohol/drug use and repeat head injury is likely to have been longer, increasing the risk of poorer self-control and impulsive behaviour. In contrast, it may be that juveniles differ by being at a transitional phase in maturation. Many juveniles desist from offending in their mid to late twenties, but some become life-course persistent offenders [3, 59]. This difference in outcome has been ascribed to developmental history, including schooling and family interactions and ongoing factors such as drug misuse, neuropsychological deficits, risk taking behaviour, antisocial attitudes, having peers who are offenders and unemployment [4, 59]. If so, juvenile prisoners may be at a 'crossroads' in their life-course, with those who have a history of SHI being at risk of further head injury and at greater risk of transition to persistent adult offending including crimes of severe violence. Our findings are consistent with this given that those with SHI have poorer self-control and higher levels of distress, but with group effects for violent offending not yet evident.

This study is limited by the absence of follow-up data which could further explore the hypothesis of juveniles with SHI being at a greater risk of recidivism. The high prevalence of SHI and problematic substance use contributed to multivariable models estimating some effects with low precision and correspondingly wide 95% CIs. The data on other CNS history relied on participants having knowledge of early life events or diagnoses. More generally, much data was collected by self-report introducing a possibility of error in recall. The Glasgow Outcome Scales have been extensively used in retrospective assessment of disability including in studies on late outcome [60], with work substantially based on single incident head injury. There might be greater risk of error in assessing disability when assessing change in function following repeated mild head injuries that occur over extended periods of time and that might have cumulative effects. Techniques to minimise potential sources of error were adopted [S1 File] and with respect to assessment of head injury, offenders do not always attend hospital when injured [61] and medical records may not reflect the high frequency of repeated, mild injuries found. Hence, self-report using a validated method is necessary. Compliance with the assessment was good. The assessment was completed in one session in about 90% of participants. The remainder required two sessions; around half of these were because of prison administrative reasons such as lock up, classes or lunch and half for participant reasons such as fatigue or restlessness. None who required a second session refused to attend.

As one in three juvenile prisoners in the UK are reconvicted within 12 months of release [62, 63] there is an imperative to offer intervention and support to reduce risks of recidivism and potentially of a transition to lifelong offending. This study indicates that juvenile prisoners with SHI show greater psychological distress and signs of poorer behavioural control and suggests they are at greater risk of further SHI, associated disability and continuing to offend. Arguably, therefore there is a need for holistic intervention that includes education about brain injury [12], support in the community and that takes account of the multiple morbidity found in this vulnerable group to facilitate lifestyle changes and provide opportunities for sustaining productive employment.

## Supporting information

**S1 File. This file contains background supporting information tables sand figures.**
(DOCX)

**S2 File. This file contains supplementary tables.**
(DOCX)

**S3 File. This file contains supplementary figures.**
(DOCX)

## Author Contributions

**Conceptualization:** T. M. McMillan.

**Data curation:** Julia McVean, Hira Aslam, Sarah J. E. Barry.

**Formal analysis:** Sarah J. E. Barry.

**Funding acquisition:** T. M. McMillan.

**Investigation:** T. M. McMillan, Julia McVean, Hira Aslam.

**Methodology:** T. M. McMillan.

**Project administration:** T. M. McMillan, Julia McVean, Hira Aslam.

**Resources:** T. M. McMillan.

**Supervision:** T. M. McMillan.

**Validation:** Sarah J. E. Barry.

**Writing – original draft:** T. M. McMillan.

**Writing – review & editing:** T. M. McMillan, Julia McVean, Hira Aslam, Sarah J. E. Barry.

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
