## [Decision Letter · Decision Letter 0]

9 Nov 2022

PONE-D-22-23246

Associations between Significant Head Injury  in Male Juveniles in Prison in Scotland UK and Cognitive Function, Disability and Crime: A Cross sectional study

PLOS ONE

Dear Dr. McMillan,

Thank you for submitting your manuscript to PLOS ONE. After careful consideration, we feel that it has merit but does not fully meet PLOS ONE’s publication criteria as it currently stands. Therefore, we invite you to submit a revised version of the manuscript that addresses the points raised during the review process.

Please see the comments from two reviewers below. Reviewer 1 has provided several suggestions, whilst reviewer 2 has only made some very minor suggestions.

In addition to the reviewer requests, please also ensure that you include the following information on resubmission:

a) A complete ethics statement in the Methods section, including the names of both approving bodies and any approval numbers

b) Information on data access restrictions, such as where and how other interested researchers are able to apply to the ethics committee or data access committee. Please include this information in the Data availability statement in the online submission form

We look forward to receiving your revised manuscript.

Kind regards,

Hanna Landenmark

Staff Editor

PLOS ONE

Journal Requirements:

Reviewers' comments:

Reviewer's Responses to Questions

**Comments to the Author**

1. Is the manuscript technically sound, and do the data support the conclusions?

Reviewer #1: Yes

Reviewer #2: Yes

2. Has the statistical analysis been performed appropriately and rigorously? 

Reviewer #1: Yes

Reviewer #2: Yes

3. Have the authors made all data underlying the findings in their manuscript fully available?

Reviewer #1: Yes

Reviewer #2: Yes

4. Is the manuscript presented in an intelligible fashion and written in standard English?

Reviewer #1: Yes

Reviewer #2: Yes

5. Review Comments to the Author

Reviewer #1: This is a well written study presenting the results of original research that has not, to my best knowledge, been published elsewhere. All other five criteria for publication on PLOS ONE are met, although I would suggest a few minor revisions to help improve reproducibility and clarity.

It would be useful to describe the procedure in a little more detail. In particular, on page six, it is mentioned that the assessments “were carried out by a final year doctorate in clinical psychology trainee and an experienced researcher. Does this mean that both assessors were present for each individual assessment, or each assessor worked with different participants. If the latter, could the authors describe what measures were taken to ensure reliability (e. g. training, inter-rater reliability procedures and estimates)?

For interest, what proportion of participants required an assessment over two sessions? This aspect can have practical implications when carrying out studies like these in “live” environments, and having an understanding of how frequent, or not, more resources are required may help other researchers better plan future studies.

Is the median number of head injuries of five reported on page 11 for the whole SHI group, or only for the 23 who had more than two injuries? If the latter, could the authors also report the median number of injuries within the SHI group overall?

Again, for interest, what was the question used to obtain information about experiences of anoxia due to strangulation?

On page 14, the authors report on the history of physical and sexual abuse, including “adult sexual abuse”. Although the interpretation would not necessarily change, should the denominator for those percentages not be calculated out of the total number of adults in the sample (i. e. only those aged ≥ 18), rather than the full sample?

Please provide more detail about how the attribution of disability on the GODS was achieved. Given the characteristics of the injuries in the SHI group, and more generally in this population (i. e. multiple, mild, and often acquired over an extended period of time), could the authors comment on the possible limitation in terms of reliability and validity of these attributions of disability specifically to head injuries?

Please revise table 4, page 17. The percentages shown on column three for the “Disability from SHI” data do not match with the total N shown on the top of that column (i. e. 70/103 is 68%, not 85%, etc.), and the same results are reported twice (on column three and column four). Should the percentages for this sub-sample not be calculated out of the total of people with head injuries (N = 82) and then out of the total of those who specifically attributed disabilities to head injury (i. e. N = 11), rather than the overall sample?

Could the authors calculate what the scores on the DEX for the 18 participants who were not rated by prison officers were, and explore whether these are comparable to the rest of the sample (in other words, is there a chance that prison officers happened not to provide ratings for those scoring themselves higher on this measure)?

It may be useful to add one or two citations about the limitations of using structured assessments and of assessing people with brain injuries and dysexecutive difficulties in structured environments (p. 23, second paragraph), as this would help readers less familiarised with the literature in this field to get more acquainted with the evidence-base for this challenge.

Minor edits

p. 7 – Expand the GODS acronym at first use.

p. 8 – Consider replacing “on more than two occasions” with “on three or more occasions” for clarity and consistency.

p. 10 and throughout – sometimes proportions are reported percentage first, then number ratios, other times the other way around (e. g. p. 10). Please use the same order throughout for simplicity.

pp. 11-12 – Consider referring to “head injuries” (rather than “head injury”) when describing multiple or repeated occurrences of injury.

p. 18 – Please ensure that all p-values are reported consistently. For example, the p-value for the difference between prison officers’ reports on the DEX by group is missing.

Reviewer #2: this is a welcome and helpful addition to the literature. a strong study, with good measures, of brain trauma in young offenders. the findings were consistent and enhance previous work. The study adds that brain injury is linked to greater issues in anxiety and other issues (eg abuse/neglect) and ongoing issues in self-regulation. these are important findings for better services around such young people. there is a very good discussions, which is timely, on such young people being at a crossroads, before greater violence becomes an issue.

the study is well conceived, well run, and has important findings.

In future it would be good to try and look more at the gradings of Mild TBI - so we get a better look at "dosage" of TBI.

6. PLOS authors have the option to publish the peer review history of their article (what does this mean?). If published, this will include your full peer review and any attached files.

Reviewer #1: No

Reviewer #2: **Yes: **Prof W. Huw Williams

---

## [Author Response · Author response to Decision Letter 0]

31 Jan 2023

Response to Reviewers

PONE-D-22-23246

Associations between Significant Head Injury in Male Juveniles in Prison in Scotland UK and Cognitive Function, Disability and Crime: A Cross sectional study

Editors Comments

In addition to the reviewer requests, please also ensure that you include the following information on resubmission:

a) A complete ethics statement in the Methods section, including the names of both approving bodies and any approval numbers

We have added to the approvals section on page 10. 

b) Information on data access restrictions, such as where and how other interested researchers are able to apply to the ethics committee or data access committee. Please include this information in the Data availability statement in the online submission form

We have added to the online data availability statement

Reviewer #1: This is a well written study presenting the results of original research that has not, to my best knowledge, been published elsewhere. All other five criteria for publication on PLOS ONE are met, although I would suggest a few minor revisions to help improve reproducibility and clarity.

It would be useful to describe the procedure in a little more detail. In particular, on page six, it is mentioned that the assessments “were carried out by a final year doctorate in clinical psychology trainee and an experienced researcher. Does this mean that both assessors were present for each individual assessment, or each assessor worked with different participants. If the latter, could the authors describe what measures were taken to ensure reliability (e. g. training, inter-rater reliability procedures and estimates)?

Changes have been made on page 6 and information on assessor training in the Supplement on page 26.

For interest, what proportion of participants required an assessment over two sessions? This aspect can have practical implications when carrying out studies like these in “live” environments, and having an understanding of how frequent, or not, more resources are required may help other researchers better plan future studies.

We do not have an exact figure. It was around 10%. We have added two sentences in the Discussion p 23/24.

Is the median number of head injuries of five reported on page 11 for the whole SHI group, or only for the 23 who had more than two injuries? If the latter, could the authors also report the median number of injuries within the SHI group overall?

It is for the 23; we think this is clearly indicated in the text already. We have added information on the median number in the SHI group overall.

Again, for interest, what was the question used to obtain information about experiences of anoxia due to strangulation?

This was derived in two ways. From responses on the OSU-TBI regarding loss of consciousness -when a participant indicated that this had occurred they were asked for information about cause. Second the “Other CNS System Compromise questions given as part of the OSU-TBI assessment asks about history of ‘oxygen deprivation (anoxia)’. This has been added to procedure on page 7.

On page 14, the authors report on the history of physical and sexual abuse, including “adult sexual abuse”. Although the interpretation would not necessarily change, should the denominator for those percentages not be calculated out of the total number of adults in the sample (i. e. only those aged ≥ 18), rather than the full sample?

We have altered the text to read ‘occurring when aged 16 or over’ rather than adult on pages 14 and 16 and in the table in the supplement on page 6. We have also added further information on the age of participants on page 10. This reflects minimum age of admission to prison as a juvenile and is more easily justified given differences/complexities in definition if ‘adulthood’ internationally.

Please provide more detail about how the attribution of disability on the GODS was achieved. Given the characteristics of the injuries in the SHI group, and more generally in this population (i. e. multiple, mild, and often acquired over an extended period of time), could the authors comment on the possible limitation in terms of reliability and validity of these attributions of disability specifically to head injuries?

The assessment of whether disability was present, how severe it might be and whether attributable to head injury followed the administration process of the Glasgow Outcome Scales which have been extensively validated. The structured interview investigates whether there is disability and if so whether this is attributable to head injury by clarifying timelines in relation to the head injury, reported sequelae, and participant attribution of symptoms to HI (if offered). There is also cross-referencing with other information, such as that obtained from the OSU-TBI to prompt during the interview and help to assist in judgements. We do not feel it appropriate to detail this in the paper as it is a general principle underpinning outcome assessments using the Glasgow Scales and follows their guidelines. We have though added to the Methods on page 8 and Discussion with regard to multiple mild injuries on pages 24/25.

Please revise table 4, page 17. The percentages shown on column three for the “Disability from SHI” data do not match with the total N shown on the top of that column (i. e. 70/103 is 68%, not 85%, etc.), and the same results are reported twice (on column three and column four). Should the percentages for this sub-sample not be calculated out of the total of people with head injuries (N = 82) and then out of the total of those who specifically attributed disabilities to head injury (i. e. N = 11), rather than the overall sample?

We see this could be misleading and have deleted the ‘total’ column from table 4. The figures in the table were in fact correct.

Could the authors calculate what the scores on the DEX for the 18 participants who were not rated by prison officers were, and explore whether these are comparable to the rest of the sample (in other words, is there a chance that prison officers happened not to provide ratings for those scoring themselves higher on this measure)?

A sentence has been added at the end of the cognitive function section on page 18.

It may be useful to add one or two citations about the limitations of using structured assessments and of assessing people with brain injuries and dysexecutive difficulties in structured environments (p. 23, second paragraph), as this would help readers less familiarised with the literature in this field to get more acquainted with the evidence-base for this challenge.

A citation to a review has been added on page 23.

Minor edits

p. 7 – Expand the GODS acronym at first use. Done (p7 last para)

p. 8 – Consider replacing “on more than two occasions” with “on three or more occasions” for clarity and consistency. No change made -this is the description we have used in previous studies.

p. 10 and throughout – sometimes proportions are reported percentage first, then number ratios, other times the other way around (e. g. p. 10). Please use the same order throughout for simplicity. Done

pp. 11-12 – Consider referring to “head injuries” (rather than “head injury”) when describing multiple or repeated occurrences of injury. I have left this as is– not to assume that during an event there were several head injuries.

p. 18 – Please ensure that all p-values are reported consistently. For example, the p-value for the difference between prison officers’ reports on the DEX by group is missing. 

We have added this p-value and a small number of others that were missing.

Reviewer #2: this is a welcome and helpful addition to the literature. a strong study, with good measures, of brain trauma in young offenders. the findings were consistent and enhance previous work. The study adds that brain injury is linked to greater issues in anxiety and other issues (eg abuse/neglect) and ongoing issues in self-regulation. these are important findings for better services around such young people. there is a very good discussions, which is timely, on such young people being at a crossroads, before greater violence becomes an issue. the study is well conceived, well run, and has important findings.

In future it would be good to try and look more at the gradings of Mild TBI - so we get a better look at "dosage" of TBI.

Thank you to reviewer 2 for comments. No changes were required.

---

## [Decision Letter · Decision Letter 1]

3 Apr 2023

PONE-D-22-23246R1

Associations between Significant Head Injury  in Male Juveniles in Prison in Scotland UK and Cognitive Function, Disability and Crime: A Cross sectional study

PLOS ONE

Dear Dr. McMillan

Thank you for submitting your manuscript to PLOS ONE. After careful consideration, we feel that this study has real merit, and so we like to invite you to submit a revised version of the manuscript that addresses some minor points raised during the review process.

Please see the comments from the reviewer below, which I feel are clear and relatively straightforward. 

I look forward to receiving your revised manuscript.

Kind regards,

Coral Dando, Ph.D.

Academic Editor

PLOS ONE

Journal Requirements:

Reviewers' comments:

Reviewer's Responses to Questions

**Comments to the Author**

1. If the authors have adequately addressed your comments raised in a previous round of review and you feel that this manuscript is now acceptable for publication, you may indicate that here to bypass the “Comments to the Author” section, enter your conflict of interest statement in the “Confidential to Editor” section, and submit your "Accept" recommendation.

Reviewer #1: All comments have been addressed

Reviewer #3: (No Response)

2. Is the manuscript technically sound, and do the data support the conclusions?

Reviewer #1: Yes

Reviewer #3: Yes

3. Has the statistical analysis been performed appropriately and rigorously? 

Reviewer #1: Yes

Reviewer #3: Yes

4. Have the authors made all data underlying the findings in their manuscript fully available?

Reviewer #1: Yes

Reviewer #3: No

5. Is the manuscript presented in an intelligible fashion and written in standard English?

Reviewer #1: Yes

Reviewer #3: Yes

6. Review Comments to the Author

Reviewer #1: Thank you to the authors for the revisions. In my view the manuscript can be accepted for publication.

Reviewer #3: This is a well-written and comprehensive study examining associations between head injury, cognitive function, and behavior problems (among other interesting associations) in a sample of male juvenile offenders currently incarcerated in Scotland. I believe the study makes a meaningful contribution to the existing literature and the authors have done a good job addressing the comments from reviewers. The resulting revised manuscript has been improved and, as a new reviewer, I do not have any additional major areas of concern. With that said, I do have two very minor suggestions that I believe would further improve the paper and should not require much effort.

1. I appreciate the brief introduction to the paper and I think the authors hit all the necessary key points, but I do think that they do not provide enough detail surrounding previous studies examining not only the association between TBI and cognitive function/behavior problems, but also the underlying mechanisms that underlie such associations. This list is not exhaustive (and I’m not asking the authors to perform a comprehensive literature review by any means) but here are a few studies that may be relevant and can be collectively summarized for interested readers who may want to dig deeper into this topic:

Sariaslan, A., Lichtenstein, P., Larsson, H., & Fazel, S. (2016). Triggers for violent criminality in patients with psychotic disorders. JAMA Psychiatry, 73(8), 796–803. https://doi.org/10.1001/jamapsychiatry.2016.1349

Sariaslan, A., Sharp, D. J., D’Onofrio, B. M., Larsson, H., & Fazel, S. (2016). Long-Term Outcomes Associated with Traumatic Brain Injury in Childhood and Adolescence: A Nationwide Swedish Cohort Study of a Wide Range of Medical and Social Outcomes. PLoS Medicine, 13(8), 15–19. https://doi.org/10.1371/journal.pmed.1002103

Schwartz, J. A. (2021). A Longitudinal Assessment of Head Injuries as a Source of Acquired Neuropsychological Deficits and the Implications for Criminal Persistence. Justice Quarterly, 38(2), 196–223. https://doi.org/10.1080/07418825.2019.1599044

Schwartz, J. A., Connolly, E. J., & Brauer, J. R. (2017). Head Injuries and Changes in Delinquency from Adolescence to Emerging Adulthood: The Importance of Self-control as a Mediating Influence. Journal of Research in Crime and Delinquency, 54(6), 869–901. https://doi.org/10.1177/0022427817710287

Schwartz, J. A., Connolly, E. J., & Valgardson, B. A. (2018). An evaluation of the directional relationship between head injuries and subsequent changes in impulse control and delinquency in a sample of previously adjudicated males. Journal of Criminal Justice, 56(August), 70–80. https://doi.org/10.1016/j.jcrimjus.2017.08.004

Schwartz, J. A., Wright, E. M., Spohn, R., Campagna, M. F., Steiner, B., & Epinger, E. (2022). Changes in Jail Admissions Before and After Traumatic Brain Injury. Journal of Quantitative Criminology, 38, 1033–1056. https://doi.org/10.1007/s10940-021-09524-7

Just to be clear, I’m not suggesting the authors should dramatically expand their literature review, but just a couple of additional sentences aimed at summarizing the existing research in this area would be beneficial.

2. Can the authors provide any additional information pertaining to how similar the demographc characteristics of their sample are compared to the overall population of juveniles at the correctional facility examined?

7. PLOS authors have the option to publish the peer review history of their article (what does this mean?). If published, this will include your full peer review and any attached files.

Reviewer #1: No

Reviewer #3: No

---

## [Author Response · Author response to Decision Letter 1]

17 Apr 2023

Response to Reviewers Comments April 2023

PONE-D-22-23246

Associations between Significant Head Injury in Male Juveniles in Prison in Scotland UK and Cognitive Function, Disability and Crime: A Cross sectional study

Editors Comments

No additional changes requested

Reviewer #1: This reviewer was happy with revisions made at the last submission. No further changes required.

Reviewer #2: This is a new reviewer -who was enthusiastic about the paper and requested two minor changes.

1.I appreciate the brief introduction to the paper and I think the authors hit all the necessary key points, but I do think that they do not provide enough detail surrounding previous studies examining not only the association between TBI and cognitive function/behavior problems, but also the underlying mechanisms that underlie such associations. This list is not exhaustive (and I’m not asking the authors to perform a comprehensive literature review by any means) but here are a few studies that may be relevant and can be collectively summarized for interested readers who may want to dig deeper into this topic:

Sariaslan, A., Lichtenstein, P., Larsson, H., & Fazel, S. (2016). Triggers for violent criminality in patients with psychotic disorders. JAMA Psychiatry, 73(8), 796–803. https://doi.org/10.1001/jamapsychiatry.2016.1349

Sariaslan, A., Sharp, D. J., D’Onofrio, B. M., Larsson, H., & Fazel, S. (2016). Long-Term Outcomes Associated with Traumatic Brain Injury in Childhood and Adolescence: A Nationwide Swedish Cohort Study of a Wide Range of Medical and Social Outcomes. PLoS Medicine, 13(8), 15–19. https://doi.org/10.1371/journal.pmed.1002103

Schwartz, J. A. (2021). A Longitudinal Assessment of Head Injuries as a Source of Acquired Neuropsychological Deficits and the Implications for Criminal Persistence. Justice Quarterly, 38(2), 196–223. https://doi.org/10.1080/07418825.2019.1599044

Schwartz, J. A., Connolly, E. J., & Brauer, J. R. (2017). Head Injuries and Changes in Delinquency from Adolescence to Emerging Adulthood: The Importance of Self-control as a Mediating Influence. Journal of Research in Crime and Delinquency, 54(6), 869–901. https://doi.org/10.1177/0022427817710287

Schwartz, J. A., Connolly, E. J., & Valgardson, B. A. (2018). An evaluation of the directional relationship between head injuries and subsequent changes in impulse control and delinquency in a sample of previously adjudicated males. Journal of Criminal Justice, 56(August), 70–80. https://doi.org/10.1016/j.jcrimjus.2017.08.004

Schwartz, J. A., Wright, E. M., Spohn, R., Campagna, M. F., Steiner, B., & Epinger, E. (2022). Changes in Jail Admissions Before and After Traumatic Brain Injury. Journal of Quantitative Criminology, 38, 1033–1056. https://doi.org/10.1007/s10940-021-09524-7

Just to be clear, I’m not suggesting the authors should dramatically expand their literature review, but just a couple of additional sentences aimed at summarizing the existing research in this area would be beneficial.

We have added a couple of sentences in the Introduction in line with the reviewer’s suggestion. We had also cited reviews here for the interested reader, which are more wide ranging (eg Williams et al 2018) and allow the reader easy entry to dig into further reading but have also added some further references as suggested. 

2.Can the authors provide any additional information pertaining to how similar the demographc characteristics of their sample are compared to the overall population of juveniles at the correctional facility examined?

We agree with the reviewer that this is an important issue and worth providing more detail. In the Methods section on page 5 we indicated that 98% of juvenile males in prison in Scotland are resident in HMYOI Polmont and are therefore likely to be representative of this population in Scotland. We have added more information comparing demographic data from Scottish Government statistics on juvenile males in prison on page 10 and in table 1 and have added a more detailed table on page 27 of the supplement with links to the relevant government database. 

Reviewer #3: No changes requested.

---

## [Editor Report · Decision Letter 2]

4 Jun 2023

Associations between Significant Head Injury  in Male Juveniles in Prison in Scotland UK and Cognitive Function, Disability and Crime: A Cross sectional study

PONE-D-22-23246R2

Dear Dr. McMillan,

We’re pleased to inform you that your manuscript has been judged scientifically suitable for publication and will be formally accepted for publication once it meets all outstanding technical requirements.

Kind regards,

Prof. Coral Dando, Ph.D.

Academic Editor

PLOS ONE
---

## [Editor Report · Acceptance letter]

8 Jun 2023

PONE-D-22-23246R2 

Associations between Significant Head Injury  in Male Juveniles in Prison in Scotland UK and Cognitive Function, Disability and Crime: A Cross sectional study 

Dear Dr. McMillan:

I'm pleased to inform you that your manuscript has been deemed suitable for publication in PLOS ONE. Congratulations! Your manuscript is now with our production department. 

Kind regards, 

on behalf of

Profesor Coral Dando 

Academic Editor

PLOS ONE